# Interaction of Smoking and Lead Exposure among Carriers of Genetic Variants Associated with a Higher Level of Oxidative Stress Indicators

**DOI:** 10.3390/ijerph18168325

**Published:** 2021-08-06

**Authors:** Kuo-Jung Ho, Tzu-Hua Chen, Chen-Cheng Yang, Yao-Chung Chuang, Hung-Yi Chuang

**Affiliations:** 1Department of Public Health, College of Health Sciences, Kaohsiung Medical University, Kaohsiung City 807, Taiwan; u103571002@kmu.edu.tw; 2Department of Family Medicine, Kaohsiung Municipal Ta-Tung Hospital, Kaohsiung Medical University, Kaohsiung City 807, Taiwan; 980264@kmuh.org.tw; 3Department of Occupational and Environmental Medicine, Kaohsiung Municipal Siaogang Hospital, Kaohsiung Medical University, Kaohsiung City 807, Taiwan; 980309@kmuh.org.tw; 4Institute for Translation Research in Biomedicine, Kaohsiung Chang Gung Memorial Hospital, Kaohsiung City 833, Taiwan; ycchuang@adm.cgmh.org.tw; 5Department of Occupational and Environmental Medicine, Kaohsiung Medical University Hospital, Kaohsiung City 807, Taiwan; 6Department of Public Health and Environmental Medicine, College of Medicine, Research Center for Environmental Medicine, Kaohsiung Medical University, Kaohsiung City 807, Taiwan

**Keywords:** lead, smoke, TBARS, OxLDL, GPx-1, CYBA, single nucleotide polymorphisms

## Abstract

Smoking and lead (Pb) exposure increased oxidative stress in human body, and people with some gene variants may be susceptible to Pb and smoking via oxidative stress. The aim of this study is to evaluate oxidative stress by measuring thiobarbituric acid reactive substances (TBARS) and the relationship of lipid peroxidation markers in Pb workers with different gene polymorphisms (rs4673 and rs1050450) in both smokers and nonsmokers. Blood samples were collected from 267 Pb workers who received their annual health examination in the Kaohsiung Medical University Hospital. Glutathione peroxidase 1 (GPx-1) rs1050450 and cytochrome B-245 Alpha Chain (CYBA) rs4673 single-nucleotide polymorphisms (SNP) were analyzed by specific primer-probes using Real-Time PCR methods. The interaction between blood Pb and smoking increased serum levels of TBARS and the ratio of oxidative low-density lipoprotein and low-density lipoprotein (oxLDL/LDL). Analysis of workers with rs1050450 SNPs showed higher blood Pb levels in the workers with CC genotype than those with CT genotype. Smokers had significantly higher blood Pb, alanine transaminase (ALT), TBARS, and OxLDL levels than nonsmokers. TBARS increased 0.009 nmol/mL when blood Pb increased one µg/dL in smokers compared to nonsmokers. The ratio of OxLDL/LDL increased 0.223 when blood Pb increased one µg/dL in smokers compared to nonsmokers. TBARS levels and the ratio of OxLDL/LDL were positively correlated and interacted between blood Pb and smoking after the adjustment of confounders, suggesting that smoking cessation is an important issue in the Pb-exposed working environment.

## 1. Introduction

Lead (Pb) is a well-known ubiquitous environmental pollutant as well as one of the most commonly used metals in industries, especially for battery manufacturing and repair, paint manufacturing, ship demolition and construction, radiation shield manufacturing, and metal soldering [1,2,3]. Adverse health effects of Pb have been reported in many human organs and systems, including cardiovascular, renal, hematologic, and nervous systems. Among them, Pb-induced oxidative stress impairs cell components, such as DNA, protein, and lipids through the generation of reactive oxidative species (ROS) [4,5].

Several antioxidant enzymes and molecules have been used for the measurement of the severity of Pb-induced oxidative stress in many human and animal studies. The most commonly used include glutathione (GSH), glutathione disulfide (GSSG), glutathione peroxidase (GPx), superoxide dismutase (SOD), catalase (CAT), and malondialdehyde (MDA) [5,6]. These oxidative stress markers present the severity of lipid peroxidation, DNA damage, and depletion of cell antioxidant defense systems, which might be indicators of Pb-induced adverse health effects [7].

Oxidative stress has been reported as a major mechanism of Pb-induced toxicity [8]. nicotinamide adenine dinucleotide/nicotinamide adenine dinucleotide phosphate (NADH/NADPH) oxidase is a key enzyme of superoxide production in the vasculature, smooth muscle cells, phagocytes, and mesangial cells [9,10], and is also implicated in a wide variety of human diseases including metabolic syndrome [11], hypertension [12], diabetes [13], renal disease [14], atherosclerosis [15], and cerebrovascular disease [16]. The cytochrome B-245 Alpha Chain (CYBA) gene (p22phox) is an essential component of NADPH oxidase located on 16q24. One of the most common single-nucleotide polymorphisms (SNP) of the CYBA gene is C242T (rs4673), resulting in a non-conservative substitution of histidine for tyrosine at codon 72 and an alteration of NAD(*p*)H activity by disrupting the heme-binding site [10]. The C242T polymorphism has been demonstrated to be related to multiple inflammatory diseases [17], metabolic disorders [18], and coronary artery disease [19].

In addition, glutathione peroxidase (GPX) is one of the antioxidant systems involved in the defense of ROS [20]. GPX1 is the most abundant enzyme in the GPX isoenzyme family [21]. GPX1 is a selenoprotein containing two exons within a 1.42-kb region located at chromosome 3p21 with antioxidant and anti-inflammatory functions [22,23].

Based on the dbSNP records (www.ncbi.nlm.nih.gov/snp, accessed on 8 May 2021), the polymorphic sites are widely distributed within the introns, exons, and UTR regions of the GPx1 gene so that they may affect the GPx1 function [23,24,25]. The rs1050450 (Pro198Leu) polymorphism is a site located within the GPx1 C-terminal region. The rs1050450 variant might be associated with cancer risk [26], might increase the risk of peripheral neuropathy in diabetes group [27], and has shown a higher prevalence of metabolic syndrome and cardiovascular diseases [24,28,29].

Low-density lipoprotein (LDL) is composed of cholesteryl ester, phospholipids, free cholesterol, triglycerides, and apolipoprotein B100 (apoB100) [30,31]. Oxidized low-density lipoprotein (OxLDL) is a complex particle derived from circulating LDL that contains lipid peroxidation and an apolipoprotein B modification, mostly from LDL, which could be used for detection of lipid oxidation [30,32]. Also, OxLDL triggered the formation of foam cells which induced the development of atherosclerosis by promoting monocyte-derived macrophages in the arterial wall and the intracellular accumulation of cholesteryl esters in these cells, which could be an indicator for evaluation of the cardiovascular disease [33].

Additionally, thiobarbituric acid reactive substances (TBARS) are also detectors for screening and monitoring lipid peroxidation [34,35,36,37]. Higher TBARS levels might increase the risk of cardiovascular diseases [38]. Schisterman EF et al. found that TBARS were the best discriminating biomarkers among reduced glutathione (GSH), Trolox equivalent antioxidant capacity (TEAC), high--density lipoprotein (HDL), uric acid, and Glutathione hydroxyperoxides peroxidase (GSHPx) when individually evaluated for coronary heart disease cases [39].

Cigarette smoking has been proved to be a source of oxidative stress [40,41]. Oxidative stress markers for smokers including F2-isoprostanes, F4-neuroprostanes, hydroxyeicosatetraenoic acid products (HETEs), 7-ketocholesterol, and 24- and 27-hydroxycholesterol were previously measured [40]. Both TBARS and OxLDL were elevated in smokers [7]. TBARS seem to be positively associated with smoking status [42].

Thus, Pb exposure and smoking impair human health by inducing oxidative damage. No investigation of the interaction of smoking and workers with Pb exposure has been performed. The aim of our research is to analyze the oxidative stress by measuring TBARS and the relationship of lipid peroxidation markers in workers with Pb exposure with different gene polymorphisms (rs4673 and rs1050450) in both smokers and nonsmokers.

## 2. Materials and Methods

### 2.1. Study Population and Health Examination

In order to evaluate the relationship between Pb exposure and oxidative damage, we performed a cross-sectional study of 267 Pb workers who received their annual health examination in Kaohsiung Medical University Hospital. The exclusion criteria included workers who were cancer patients and taking hypercholesterolemia medicine. Anthropometric measurements including body mass index (BMI) were checked. Blood samples were obtained including lipid profile (total cholesterol, triglyceride, low-density lipoprotein [LDL], high-density lipoprotein [HDL]), fasting sugar (AC), liver function test (alanine transaminase, ALT), renal function test (serum creatinine, Cr), and oxidative stress markers (OxLDL, TBARS). The blood Pb level was measured by Graphite Furnace Atomic Absorption Spectrometry (GFAAS). All the samples were measured in the Central Laboratory in Kaohsiung Medical University Hospital. The study was approved by the IRB of Kaohsiung Medical University Hospital (KMUHIRB-E(I)-20190034) for experimentation with human subjects. In addition, a questionnaire with information on job title, medical history, working history, personal history of alcohol and cigarette consumption was collected from each participant after a full explanation of the study and informed consent signed.

### 2.2. Genotyping Methods for Glutathione Peroxidase-1 (GPx-1) Gene (rs1050450) and Cytochrome b Light Chain (CYBA) Gene (rs4673)

Genomic DNA was extracted from peripheral blood using FlexiGene (Qiagen, Hilden, Germany) following the manufacturer’s protocol. Each real-time PCR was performed with a 10 uL reaction volume mix fluid, containing 5 µL Genotyping Master Mix (Applied Biosystems, Waltham, MA, USA), 3.87 uL distilled water, 1 uL DNA fluid (10 ng/uL) and 0.25 uL primer-probe. Amplification reactions were performed using the following program for total 45 cycles: 50 °C for 2 min; 92 °C for 10 min; 95 °C for 15 s; 60 °C for 1 min. We used TaqMan Allelic Discrimination Assays (Applied Biosystems, Foster City, CA, USA) to genotype the SNPs, and the results were read with a 7300 Real-time PCR System (Life Technologies Corp., Carlsbad, CA, USA). The fluorescence level was measured with an Applied Biosystems StepOne Real-Time PCR System (Applied Biosystems, Waltham, USA). The allele frequencies were determined using ABI SDS software. Genotyping was repeated on a random 10% sample to confirm the results of the original run by the laboratory technicians, who were blinded to the original results. The estimated genotyping error rate was less than 1%.

### 2.3. Measurement of TBARS and Oxidized LDL

Lipid peroxidation as an indicator of oxidative damage was determined by measuring the plasma concentration of TBARS using the method of Ohkawa et al. [43]. A standard curve of TBARS was obtained by hydrolysis of 1,1,3,3-tetraethoxypropane. Activation of the antioxidative defense in response to increased oxidative damage was evaluated by measuring the plasma level of total reduced thiols, a physiological free radical scavenger. Plasma thiols were determined by direct reaction with 5,5-dithiobis (2-nitrobenzoic acid) to form 5-thio-2-nitrobenzoic acid (TNB) and then calculated from the absorbance using the extinction coefficient of TNB (A_412_ = 13,600·M^−1^ cm^−1^) [44].

Oxidized LDL concentrations were measured using a sandwich enzyme-linked immunosorbent assay (ELISA) according to the manufacturer’s instructions (Mercodia oxidized LDL ELISA; Mercodia AB, Uppsala, Sweden). In brief, diluted plasma and standards were incubated in the wells of a microtiter plate coated with murine monoclonal antioxidized LDL antibodies (mAb-4E6) for 2 h at room temperature. After washing 6 times to remove nonreactive plasma components, a peroxidase-conjugated antiapolipoprotein B antibody was added to the wells with the oxidized LDL bound to the solid phase. After a second incubation for 1 h at room temperature and a washing step to remove the unbound enzyme-labeled antibody, the bound conjugate was detected by a reaction with 3,3′,5,5′-tetramethylbenzidine (TMB). This reaction was stopped by adding 1 M H_2_SO_4_, and the optical density (OD) was measured at OD 450 nm using a microplate reader. The results were calculated using the computerized data reduction of absorbance for the standards versus the concentration using cubic spline regression [45].

### 2.4. Data Analysis

There are different criteria for the definition of obesity among adults over the world. The World Health Organization (WHO) defines the Asia criteria of overweight as BMI ≥ 23 and obese as BMI ≥ 25 [46]. Both WHO and the National Institutes of Health (NIH) have promoted BMI cutoffs of 25 for overweight and 30 for obesity [47,48]. Lifestyle and genetic differences among different race and ethnicity could explain that Chinese and other Asian groups have a higher percentage of body fat than Caucasians given the same level of BMI [49]. The Taiwan Department of Health (DOH) defined overweight as BMI ≥ 24 and obese as BMI ≥ 27 [50]. A cutoff value of 24 was used in our study according to the definition of overweight of the Taiwan DOH.

Some previous researchers have used the ratio of OxLDL–to–LDL because it might reflect the clinical relevance of oxidative stress more accurately [31,51]. This process was adopted from our research.

Statistical analysis was performed using statistical software package (SPSS 20.0, IBM, Armonk, New York, NY, USA). The continuous variables including blood Pb, age, body mass index (BMI), creatinine, ALT, AC sugar, TBARS, and OxLDL were presented as mean ± SD. Proportions were used for categorical variables such as sex, BMI < 18.5, 18.5 ≤ BMI < 24, BMI ≥ 24, Cr > 1.5, ALT > 80, AC sugar ≥ 126, rs4673, and rs1050450 SNPs. The genotype distributions for both polymorphisms were calculated using the Hardy–Weinberg equilibrium equation. The study population was divided by different genotypes of rs4673 (CC, CT, TT) and rs1050450 (CC, CT). One-way ANOVA was used to examine the differences of continuous variables and a chi-square test was used to compare categorical variables among the different types of SNPs. We used independent t tests to examine the differences of continuous variables and chi-square tests to compare categorical variables among the different types of rs1050450 because the SNP only had 2 types (CC and CT). Regression analysis was used to evaluate the association of oxidative stress markers (TBARS and OxLDL) in groups of smokers and nonsmokers, blood Pb, and SNPs after adjusting for age, sex, BMI < 18.5, BMI ≥ 24 versus 18.5 ≤ BMI < 24, Cr > 1.5 versus Cr ≤ 1.5, ALT > 80 versus ALT ≤ 80, AC sugar ≥ 126 versus AC < 126. We then analyzed the interaction terms between Pb and smoke, Pb and different SNPs on the plasma oxidative stress markers (TBARS and OxLDL/LDL) as dependent variables. A two-tailed *p*-value < 0.05 was considered significant.

## 3. Results

Of the total 267 workers, the highest blood Pb was 58.93 mcg/dL and the lowest was 0.03 mcg/dL (office worker) with a mean 12.3 mcg/dL (SD = 13.7). 193 (72.3%) workers were male. Smokers account for 30.3% of these workers. More than half of them were overweight (BMI higher than 24) according to the Taiwan DOH definition. Less than 10% of the population has abnormal renal function, liver function, and fasting sugar. Analysis of CYBA rs4673 polymorphism showed 205 workers with CC, 58 with CT, and 4 with TT types, which was consistent with Hardy-Weinberg equilibrium (*p* = 0.96). Analysis of GPx-1 rs1050450 polymorphism revealed only 236 with CC and 31 CT types, without TT type (consistent with Hardy-Weinberg equilibrium, *p* = 0.31).

Table 1 shows that smokers had significantly higher blood Pb, BMI, and ALT levels than nonsmokers. In addition, smokers reported using more alcohol. With regard to markers of oxidative stress, TBARS and OxLDL are significantly higher in smokers. Men had a significantly higher prevalence of smokers in comparison with women.

Table 2 shows the descriptive characteristics among the different types of the CYBA rs4673 and GPx-1 rs1050450 SNPs. Rs4673 have three genotypes (CC, CT, and TT). The TT genotype has the highest blood level and the lowest oxidative stress markers, however not statistically significant. Analysis of GPx-1 rs1050450 SNP showed a statistically significant difference in blood Pb that workers with CC genotypes higher than those with CT genotypes. A comparative analysis of TBARS and OxLDL found no significant difference between CC and CT genotypes.

In Table 3, we present results of multiple linear regression using the markers of oxidation (TBARS and OxLDL/LDL) as dependent variables on SNPs and blood Pb after adjustment for age, sex, BMI < 18.5, BMI ≥ 24 versus 18.5 ≤ BMI < 24, Cr > 1.5 versus Cr ≤ 1.5, ALT > 80 versus ALT ≤ 80, AC sugar ≥ 126 versus AC < 126.. We explored the interactive relationship of blood Pb, different SNPs, and smoking status from Models 1–5, which showed a significant positive interaction between blood Pb and smoke when TBARS and OxLDL/LDL were used as dependent variables. In addition, blood Pb alone is a positive variable in Models 2, 3, and 4 when OxLDL/LDL was the dependent variable.

In Figure 1a, TBARS increased 0.009 nmol/mL when blood Pb increased one mcg/dL in smokers compared to nonsmokers. In Figure 1b, the ratio of OxLDL/LDL increased 0.223 when blood Pb increased one mcg/dL in smokers compared to nonsmokers.

## 4. Discussion

The major findings of this study are that smoking combined with Pb exposure had a synergistic effect that could increase serum levels of TBARS and the ratio of OxLDL/LDL after adjustment for age, sex, abnormal BMI, ALT, creatinine, and AC sugar. The previous animal studies reported an increase of serum TBARS level in Pb-exposed rats [52,53]. Meanwhile, TBARS and oxLDL levels had a positive correlation with cigarette smoking [42,54,55]. To our knowledge, there were few studies investigating the relationship between TBARS or OxLDL/LDL and the interaction between Pb and smoke. We discovered that the interaction between Pb and smoke significantly increased TBARS levels and the ratio of OxLDL/LDL, which suggested that smoking cessation is an important issue in the Pb-exposed working environment.

A study revealed nonsmoking Type II diabetic patients with macrovascular disease had a higher OxLDL/LDL ratio, with 30% of the study participants in the upper tertiles of the OxLDL/LDL ratio having a significantly higher incidence of macrovascular diseases than those in the lower tertiles of the OxLDL/LDL ratio (18% incidence of macrovascular diseases) [31]. Jeffrey et al. found that the risk of developing coronary heart disease (CHD) was significantly higher in the highest quartile of the ratio of OxLDL/LDL than the other three quartiles in male diabetic patients. The percentage of CHD in men in the upper quartiles was 35.1%, and only 18.4% in the lower quartiles [51]. It seems that the ratio of OxLDL/LDL could be a useful indicator for evaluating the development of macrovascular disease among diabetic patients. We suggest future research with different methods, and with different criteria of selection of the study participants as the present study results could be an initial guide for exploration of the relationship between workers exposed to Pb and coronary heart disease.

Smokers exposed to Pb have higher TBARS levels in our study, which led us to assess the prevalence of cardiovascular diseases (CVD) in our group as the previous study revealed an increased risk of CVD with higher TBARS levels.

GPX1 is a selenoprotein containing two exons within a 1.42-kb region located at chromosome 3p21 with antioxidant and anti-inflammatory functions [22,23]. A transition of C to T allele of the GPX1 gene (rs1050450) means the change of amino acid from proline (Pro) to leucine (Leu) at codon 198 (Pro198Leu) [56]. The T allele in the rs1050450 locus was shown to have less antioxidant capacity than the wild-type allele C of the rs1050450 locus because of less selenium-enhanced GPX1 expression [57,58]. The previous study has shown that T allele frequency in the Asian population is less than 0.2. This is consistent with our study, which showed 0.116 [59]. Other research has revealed the incidence of T allele as between 0.41 and 0.58 in the non-Asian population, while Western Europe has a lower frequency of T allele than North America and Northern Europe [60,61,62,63,64,65,66,67]. Our study did not reveal a significant effect of rs1050450 SNPs on oxidative stress markers, TBARS, or on the ratio of OxLDL/LDL, which may be due to lack of TT genotype.

The C242T polymorphism is located in exon 4 at position 214 from the ATG codon [17]. The C242T polymorphism nucleotide transition encodes a CAC→TAC codon change, thus resulting in a non-conservative substitution of Histidine-72 with tyrosine, an alteration that may modify the heme-binding site for the stability of the CYBA gene protein [68]. This replacement is expected to reduce oxidative function and to decrease the production of ROS and oxidative stress in the vasculature [69]. A previous study revealed the relationship between rs4673 polymorphism and TBARS, which showed no significant difference between the genotypes [70]. Our study showed the same results. However, the study population of that study was confined to type 2 diabetic patients, whereas our study population was focused on Pb-exposed workers without confinement to type 2 diabetes. A broader population could be studied for further analysis and we could also use different oxidative stress markers that would better stand for a standard method of oxidative stress in vivo in the future. Takanari et al. used OxLDL in addition to TBARS for evaluation of different genotypes of rs4673. The multiple regression analysis showed slightly significantly higher concentrations of OxLDL in those with T allele, which was unexpected. In our study, considering the total concentrations of serum lipid status, we used the ratio of OxLDL to LDL for correction, which showed insignificant results between rs4673 SNPs and OxLDL/LDL. The influence of rs4673 might be covered by smoking status and BMI, both of which are strong oxidative resources.

The amendment to the Tobacco Hazards Prevention Act in 2009 showed smoking prevalence among men aged 18 and older in Taiwan 33.5% [71,72]. The smoking prevalence of our study is similar to the general population in Taiwan. The prevalence of overweight participants in our study is more than two-folds higher than in the general population, which was 23.9% according to Taiwan’s DOH criteria in 2001 [50]. Blood Pb level is the highest, while, inversely, TBARS and OxLDL levels are the lowest in rs4673 TT polymorphism, though the numbers are not statistically significant. Data on larger populations needs to be collected for further investigation of our results. US research presented that smokers had higher blood Pb compared to nonsmokers because tobacco contained Pb [73], which is consistent with our study.

The limitation of our study is that not all variants of the GPX1 gene and CYBA gene were analyzed in this study. Other functional SNPs may interact with each other, causing different outcomes. On the other hand, we only recorded the current smoking status in this study without details about the number of packs per year. The measurement of packs per year in a future study would help us identify the dose–response relationship of quantification of smoking and oxidative stress, which would provide the precise cut-off value for initiation of smoking cessation in Pb-exposed workers. In addition, the majority of our participants were male which made it difficult to look at sex effects and also that lead levels varied between smoking and nonsmoking groups. However, this limitation wildly appeared in the research consistent of heavy workers, and might not influence our conclusion.

## 5. Conclusions

TBARS levels and the ratio of OxLDL/LDL were positively correlated with the interaction between blood Pb and smoking after the adjustment of confounders, suggesting that smoking cessation is an important issue in the Pb-exposed working environment.

## Figures and Tables

**Figure 1 ijerph-18-08325-f001:**
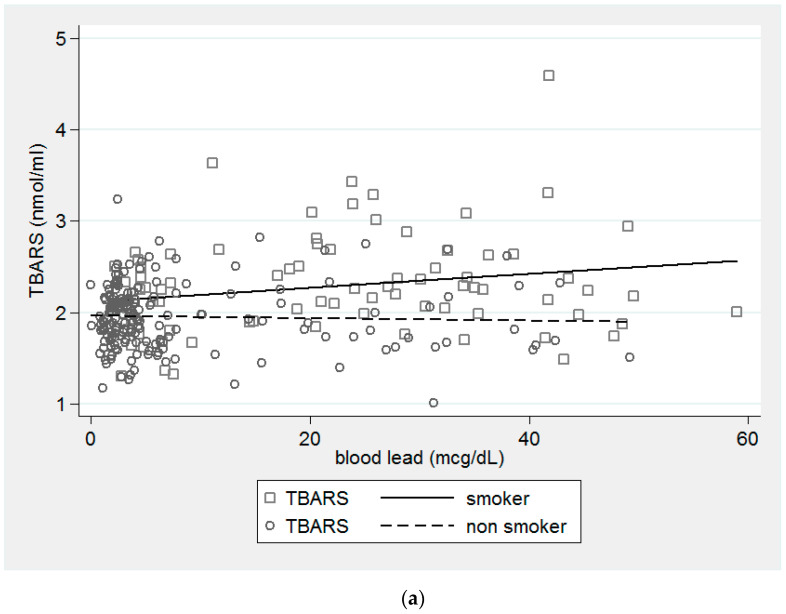
The interaction effect of smoking on the levels of TBARS and ratio of OxLDL/LDL in Pb-exposed workers. (**a**) The difference between the levels of TBARS was greater for smokers than nonsmokers in Pb-exposed workers. (*p* = 0.041). (**b**) The difference between the ratio of OxLDL/LDL was greater for smokers than for nonsmokers. (*p* = 0.044).

**Table 1 ijerph-18-08325-t001:** Descriptive data of clinical and biochemical by smoking status.

Variables	Smokers (*n* = 81)	Nonsmokers (*n* = 186)	*p* Value
Blood Pb (μg/dL)	22.11 ± 15.19	7.97 ± 10.42	<0.001 **
Age (years)	43.7 ± 9.7	41.1 ± 10.4	0.058
Sex (male)	80 (98.8%)	113 (60.8%)	<0.001 **
Alcohol (>3 times/week)	28 (42.4%)	13 (7.3%)	<0.001 **
BMI (kg/m^2^)	25.69 ± 3.52	23.88 ± 3.71	<0.001 **
BMI < 18.5	3 (3.7%)	7 (3.8%)	0.981
BMI ≥ 24	56 (69.1%)	83 (44.6%)	0.001 **
Creatinine (mg/dL)	1.22 ± 0.14	1.15 ± 0.63	0.357
ALT (U/L)	22.85 ± 14.42	18.19 ± 16.14	0.027 *
AC sugar (mg/dL)	102.92 ± 33.41	99.54 ± 31.08	0.433
TBARS (nmol/mL)	2.289 ± 0.559	1.953 ± 0.351	<0.001 **
OxLDL (mg/dL)	59.857 ± 14.567	53.76 ± 11.252	0.001 **
LDL (mg/dL)	104.621 ± 18.304	104.457 ± 11.625	0.942
CYBA gene (rs4673)			0.625
CC	60 (74.1%)	145 (78.0%)	
CT	19 (23.5%)	39 (21.0%)	
TT	2 (2.5%)	2 (1.1%)	
GPX1 gene (rs1050450)			0.157
CC	75 (92.6%)	161 (86.6%)	
CT	6 (7.4%)	25 (13.4%)	

* *p* < 0.05, ** *p* < 0.01. *p* values were calculated by independent *t* test for continuous variables and chi-square test for categorical variables. Types of CYBA (rs4673) and GPX1 (rs1050450) in both smokers and nonsmokers groups were consistent with Hardy–Weinberg equilibrium. BMI = body mass index; ALT = alanine transaminase, AC sugar = fasting sugar; TBARS = thiobarbituric acid reactive substances; OxLDL = oxidative low-density lipoprotein; LDL = low-density lipoprotein; CYBA gene = cytochrome B-245 Alpha Chain gene; GPX1 gene = Glutathione peroxidase 1 gene.

**Table 2 ijerph-18-08325-t002:** Descriptive characteristics among the different types of CYBA rs4673 and GPx-1 rs1050450 SNPs.

Variables	CYBA rs4673				GPx-1 rs1050450		
Mean ± SD or *n* (%)	CC (*n* = 205)	CT (*n* = 58)	TT (*n* = 4)	*p* Value	CC (*n* = 236)	CT (*n* = 31)	*p* Value
Blood Pb (μg/dL)	11.82 ± 13.74	13.09 ± 12.75	23.04 ± 22.31	0.234	12.87 ± 14.03	7.58 ± 9.61	0.009 *
Age (years)	42.1 ± 10.1	41.6 ± 11.2	35.6 ± 4.4	0.477	42.0 ± 10.3	40.9 ± 9.8	0.556
Sex (male)	148 (72.2%)	41 (70.7%)	4 (100%)	0.448	171 (72.5%)	22 (71.0%)	0.862
Current smoking	60 (29.3%)	19 (32.8%)	2 (50%)	0.605	75 (31.8%)	6 (19.4%)	0.157
Alcohol (>3 times/week)	32 (16.9%)	9 (17.3%)	0 (0%)	0.734	38 (17.6%)	3 (10.7%)	0.36
BMI (kg/m^2^)	24.43 ± 3.72	24.48 ± 3.91	24.48 ± 2.55	0.875	24.51 ± 3.81	23.83 ± 3.14	0.343
BMI < 18.5	8 (3.9%)	2 (3.4%)	0 (0%)	0.912	9 (3.8%)	1 (3.2%)	0.871
BMI ≥ 24	106 (51.7%)	32 (55.1%)	1 (25%)	0.494	123 (52.1%)	16 (51.6%)	0.958
Creatinine (mg/dL)	1.18 ± 0.60	1.15 ± 0.18	1.17 ± 0.53	0.954	1.18 ± 0.59	1.10 ± 0.17	0.438
ALT (U/L)	19.7 ± 15.1	20.1 ± 18.3	11.0 ± 6.3	0.624	19.9 ± 16.2	17.8 ± 11.7	0.501
AC sugar (mg/dL)	98.9 ± 25.1	107.3 ± 49.0	90.2 ± 5.4	0.192	101.3 ± 33.3	94.8 ± 14.0	0.308
TBARS(nmol/mL)	2.06 ± 0.46	2.06 ± 0.41	1.72 ± 0.27	0.332	2.053 ± 0.464	2.078 ± 0.344	0.776
OxLDL (mg/dL)	55.98 ± 13.02	55.04 ± 11.26	45.87 ± 10.74	0.265	55.64 ± 12.92	55.50 ± 10.53	0.955
LDL (mg/dL)	105.31 ± 13.53	102.54 ± 14.89	93.07 ± 16.5	0.105	104.00 ± 14.10	108.39 ± 12.23	0.105

* *p* < 0.05. *p* values were calculated by independent t test for continuous variables and chi-square test for categorical variables.

**Table 3 ijerph-18-08325-t003:** Multiple linear regression coefficients of TBARS and OxLDL/LDL in different interaction models.

Models	TBARS	OxLDL/LDL
	β (SE)	β (SE)
Model 1		
Blood Pb	−0.001 (0.003)	0.078 (0.08)
smoking	0.115 (0.101)	−0.704 (2.447)
Blood Pb x smoke	0.009 (0.005) *	0.223 (0.11) *
Model 2		
Blood Pb	0.004 (0.003)	0.193 (0.062) *
rs4673	0.003 (0.092)	0.336 (2.219)
Blood Pb x rs4673	−0.003 (0.005)	0 (0.114)
Model 3		
Blood Pb	0.004 (0.002)	0.212 (0.058) **
rs1050450	0.137 (0.114)	2.054 (2.746)
Blood Pb x rs1050450	−0.007 (0.009)	−0.317 (0.205)
Model 4		
Blood Pb	0.004 (0.002)	0.191 (0.057) **
rs4673	−0.055 (0.069)	0.115 (1.67)
rs1050450	0.051 (0.104)	−1.278 (2.514)
rs4673 x rs1050450	0.1 (0.193)	2.037 (4.66)
Model 5		
Blood Pb	0.004 (0.002)	0.192 (0.057) **
rs4673	−0.038 (0.067)	0.433 (1.619)
rs1050450	0.09 (0.098)	−0.559 (2.352)
Blood Pb x rs4673 x rs1050450	−0.003 (0.012)	−0.038 (0.285)

* *p* < 0.05, ** *p* < 0.01, β (SE): regression coefficient and standard error.All models were adjusted for age, sex, BMI < 18.5, BMI ≥ 24 versus 18.5 ≤ BMI < 24, Cr > 1.5 versus Cr ≤ 1.5, ALT > 80 versus ALT ≤ 80, AC sugar ≥ 126 versus AC < 126.

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
