# Peer review of "Interaction of Smoking and Lead Exposure among Carriers of Genetic Variants Associated with a Higher Level of Oxidative Stress Indicators"

_ijerph, 2021, doi:10.3390/ijerph18168325_

Round 1

Reviewer 1 Report

Abstract: Define all abbreviations in abstract.

introduction:

Lines 40-42 needs a citation:

Obeng-Gyasi, E. and Obeng-Gyasi, B., 2020. Chronic stress and cardiovascular disease among individuals exposed to lead: A pilot study. Diseases, 8(1), p.7.

Harari, F., Sallsten, G., Christensson, A., Petkovic, M., Hedblad, B., Forsgard, N., Melander, O., Nilsson, P.M., Borné, Y., Engström, G. and Barregard, L., 2018. Blood lead levels and decreased kidney function in a population-based cohort. American Journal of Kidney Diseases, 72(3), pp.381-389.

Rocha, A. and Trujillo, K.A., 2019. Neurotoxicity of low-level lead exposure: History, mechanisms of action, and behavioral effects in humans and preclinical models. Neurotoxicology, 73, pp.58-80.

Kianoush, S., Balali-Mood, M., Mousavi, S.R., Shakeri, M.T., Dadpour, B., Moradi, V. and Sadeghi, M., 2013. Clinical, toxicological, biochemical, and hematologic parameters in lead exposed workers of a car battery industry. Iranian journal of medical sciences, 38(1), p.30.

Methods:

Did you run a ShapiroWilk test to see how your data was distributed? Log transformation may have been warranted for some of the analysis.

Why did you not adjust for medication consumption?

Why did you not explore inflammatory biomarkers?

Discussion:

You need to compare your results with  other studies to give the reader context for the  results.

You need to discuss limitations of study design.

Author Response

Abstract: Define all abbreviations in abstract.

Response:

Thanks for the comment, we have revised the manuscript according to your suggestions.

introduction:

Lines 40-42 needs a citation:

Obeng-Gyasi, E. and Obeng-Gyasi, B., 2020. Chronic stress and cardiovascular disease among individuals exposed to lead: A pilot study.Diseases,8(1), p.7.

Harari, F., Sallsten, G., Christensson, A., Petkovic, M., Hedblad, B., Forsgard, N., Melander, O., Nilsson, P.M., Borné, Y., Engström, G. and Barregard, L., 2018. Blood lead levels and decreased kidney function in a population-based cohort. American Journal of Kidney Diseases72(3), pp.381-389.

Rocha, A. and Trujillo, K.A., 2019. Neurotoxicity of low-level lead exposure: History, mechanisms of action, and behavioral effects in humans and preclinical models. Neurotoxicology73, pp.58-80.

Kianoush, S., Balali-Mood, M., Mousavi, S.R., Shakeri, M.T., Dadpour, B., Moradi, V. and Sadeghi, M., 2013. Clinical, toxicological, biochemical, and hematologic parameters in lead exposed workers of a car battery industry. Iranian journal of medical sciences38(1), p.30.

Response:

Thanks for the comment, we have revised the manuscript including these references, lines 352-368.

Methods:

Did you run a ShapiroWilk test to see how your data was distributed? Log transformation may have been warranted for some of the analysis.

Left(right?) skewed distribution…

Response:

Thanks for the comment, we had used log transformation before; however, even log transformed, the distribution still skewed, and the regression results were similar significance; thus we remained non-transformed.

Why did you not adjust for medication consumption?

Response:

Thank you for this valuable comment. The study participants were all healthy workers. We have excluded who took hyper cholesterol medicine, please see lines 102-103.

Why did you not explore inflammatory biomarkers?

Response:

Thank you for your insightful opinion. But we did not have the data. We focused more on the smoking, metal exposure and genetic polymorphisms, and the relationship of oxidative stress in this study.

On the other hand, thanks for the comment. To explore inflammatory biomarkers may be our next interesting researches.

Discussion:

You need to compare your results with other studies to give the reader context for the results.

Response:

Thank you for your insightful opinion. We tried to compare our finding to the other researches in discussion, and cited more than 70 references. However, as we know, this is a pioneer study using TBARS and OxLDL as the biomarkers in lead-exposed population, and discussed the interactions of Pb-exposure, genotypes, and smoking associated to oxidative biomarkers, hopefully this can give some informative values to the readers.

You need to discuss limitations of study design.

Response:

Thank you for your insightful opinion. We have discussed the limitation on lines 320-326.

Reviewer 2 Report

Kuo-Jung Ho and colleagues presented here article entitled "Interaction of Smoking and Lead Exposure Increasing Oxidative Stress Indicators among Related Genetic Polymorphisms" that raises the issue of lead exposure and its more toxic effect among smokers carrying risk alleles of sequence variants in genes encoding proteins related to oxidative stress.

I find the presented study interesting and fit into the scope of the journal. Results seem to be worthy of publishing, but the article must be improved. As I'm a geneticist, I recommend checking and correcting the genetic part.

The authors should start with the title because the phrase "among Related Genetic Polymorphisms" is confusing. I think the authors meant: among carriers of genetic variants associated with a higher level of oxidative stress?

In my opinion, the abstract should be moderated because it does not explain which gene polymorphisms are being studied and why. In turn, the description of the methods is too detailed. The abstract should interest the reader and be transparent.

I suggest consulting the genetic part of the study with a geneticist because there are mistakes in the interpretation of "allele" and "genotype" in the whole article body".

T is an allele

TT is genotype

SNP is the shortcut from single nucleotide polymorphism

For example – in lines 190-192:

"CYBA  rs4673 polymorphisms showed 205 workers with CC, 58 with CT, and 4 with TT types, which was consistent with Hardy-Weinberg equilibrium (p=0.96). GPx-1 rs1050450 polymorphisms revealed only 236 with CC and 31 CT types without TT type."

The sentences should be for example:

CYBA gene rs4673 polymorphism analysis showed 205 individuals with CC, 58 with CT, and 4 with TT genotype, which was consistent with Hardy-Weinberg equilibrium (P-value=0.96). GPX1 gene rs1050450 polymorphism analysis revealed 236 carriers of CC genotype and 31 carriers of CT genotype, while TT genotype was not present.

Discussion fragment in lines 269-311 must be corrected regarding appropriate use of term allele, genotype, SNP. There is a lot of serious mistakes in this fragment.

For example:

In line 273: "wild-type allele (CC) of the rs1050450 gene" – should be wild-type allele (C) (– because CC is a genotype), of the rs1050450 polymorphism/locus in GPX1 gene.

Line 280: "rs1050450 SNPs might be related to different distributions of T alleles among different ethnicities and areas, other SNPs, environmental factors, and/or dietary intake of fruit and vegetables."

This sentence does not have a sense for me. What do You mean?

Next, the description of tables 2 and 3 in lines 208-214 must be corrected – it is confusing. Maybe something is missing in line 211?

"Table 2 shows the descriptive characteristics of the distribution of CYBA rs4673 and GPx-1 rs1050450 SNPs.. Rs4673 have three genotypes. The TT genotype has the highest blood level, and the has the lowest oxidative stress markers among these three. Table 3 shows the demographic characteristics of the distribution of There is a statistically significant difference in blood Pb, with CC genotypes higher than CT genotypes. A comparative analysis of TBARS and OxLDL found no significant difference between different genotypes."

In line 215 is table 1– it should be table 2.

Moreover, I suggest unifying the units. In tables 1 and 2, the Authors used: Blood Pb - (μg/dL), but in the text, they wrote mcg/dL. The same with P-value (there is P, p, or P value, for example, in and under tables 1 and 2). The writing style must be the same throughout the paper.

In table 3, there are: β and (SE) in the caption – these symbols/shortcuts are not explained.

In the Methods section in lines 122-123, there is:

"The amplified products were digested at 37°C or 60°C for 1 hour 122 each using 25-μL PCR-amplified product with 10 units of the restriction enzyme…" - 

Please specify what enzyme did you use and what for?

In conclusion, I consider the research presented to be valuable, but in this form, it cannot be published in the International Journal of Environmental Research and Public Health. I recommend revisions and resubmission of a revised article to the editor.

Author Response

Kuo-Jung Ho and colleagues presented here article entitled "Interaction of Smoking and Lead Exposure Increasing Oxidative Stress Indicators among Related Genetic Polymorphisms" that raises the issue of lead exposure and its more toxic effect among smokers carrying risk alleles of sequence variants in genes encoding proteins related to oxidative stress.

I find the presented study interesting and fit into the scope of the journal. Results seem to be worthy of publishing, but the article must be improved. As I'm a geneticist, I recommend checking and correcting the genetic part.

The authors should start with the title because the phrase "among Related Genetic Polymorphisms" is confusing. I think the authors meant: among carriers of genetic variants associated with a higher level of oxidative stress?

Response:

We would like to thank the reviewer for the comment. The title has changed to “Interaction of Smoking and Lead Exposure Increasing Oxidative Stress Indicators among Carriers of Genetic Variants Associated with a Higher Level of Oxidative Stress Indicators”

In my opinion, the abstract should be moderated because it does not explain which gene polymorphisms are being studied and why. In turn, the description of the methods is too detailed. The abstract should interest the reader and be transparent.

Response:

We appreciate your comments, and have rewritten the abstract to be readable.

I suggest consulting the genetic part of the study with a geneticist because there are mistakes in the interpretation of "allele" and "genotype" in the whole article body".

T is an allele

TT is genotype

SNP is the shortcut from single nucleotide polymorphism

For example – in lines 190-192:

"CYBA  rs4673 polymorphisms showed 205 workers with CC, 58 with CT, and 4 with TT types, which was consistent with Hardy-Weinberg equilibrium (p=0.96). GPx-1 rs1050450 polymorphisms revealed only 236 with CC and 31 CT types without TT type."

The sentences should be for example:

CYBA gene rs4673 polymorphism analysis showed 205 individuals with CC, 58 with CT, and 4 with TT genotype, which was consistent with Hardy-Weinberg equilibrium (P-value=0.96). GPX1 gene rs1050450 polymorphism analysis revealed 236 carriers of CC genotype and 31 carriers of CT genotype, while TT genotype was not present.

Response:

Thanks for the comment. We agree with the reviewer as the information can be confusing. And the manuscript was re-written, according to your suggestion, as “Analysis of CYBA rs4673 polymorphism showed 205 workers with CC, 58 with CT, and 4 with TT types, which was consistent with Hardy-Weinberg equilibrium (p=0.96). Analysis of GPx-1 rs1050450 polymorphisms revealed only 236 with CC and 31 CT types, without TT type (consistent with Hardy-Weinberg equilibrium, p=0.31).” …… lines 197-201.

Discussion fragment in lines 269-311 must be corrected regarding appropriate use of term allele, genotype, SNP. There is a lot of serious mistakes in this fragment.

For example:

In line 273: "wild-type allele (CC) of the rs1050450 gene" – should be wild-type allele (C) (– because CC is a genotype), of the rs1050450 polymorphism/locus in GPX1 gene.

Response:

We thank the reviewer for suggesting a better expression. We’ve revised as your advice. Lines 283-286 “The T allele in the rs1050450 gene was shown to have less antioxidant capacity than the wild-type allele C of the rs1050450 gene because of less selenium-enhanced GPX1 expression [57,58].”

This paragraph was carefully re-written to avoid the mistakes that the respectful reviewer mentioned.

Line 280: "rs1050450 SNPs might be related to different distributions of T alleles among different ethnicities and areas, other SNPs, environmental factors, and/or dietary intake of fruit and vegetables."

This sentence does not have a sense for me. What do You mean?

Response:

We appreciate your comments, and this is a fragment of draft. We have deleted, Line 280 of the previous manuscript.

Next, the description of tables 2 and 3 in lines 208-214 must be corrected – it is confusing. Maybe something is missing in line 211?

"Table 2 shows the descriptive characteristics of the distribution of CYBA rs4673 and GPx-1 rs1050450 SNPs.. Rs4673 have three genotypes. The TT genotype has the highest blood level, and the has the lowest oxidative stress markers among these three. Table 3 shows the demographic characteristics of the distribution of There is a statistically significant difference in blood Pb, with CC genotypes higher than CT genotypes. A comparative analysis of TBARS and OxLDL found no significant difference between different genotypes."

Response:

We are sorry that revised fragment of draft editing error, and thank the reviewer for telling us the mistake. The paragraph was re-written as “Table 2 shows the descriptive characteristics among the different types of the dis-tribution of CYBA rs4673 and GPx-1 rs1050450 SNPs.. Rs4673 have three genotypes (CC, CT, and TT). The TT genotype has the highest blood level and the has the lowest oxidative stress markers among these three, however not statistically significant. Table 3 shows the demographic characteristics of the distribution of There isAnalysis of GPx-1 rs1050450 SNP showed a statistically significant difference in blood Pb that, with workers with CC genotypes higher than those with CT genotypes. A comparative analysis of TBARS and OxLDL found no significant difference between different CC and CT genotypes.” --- lines 216-223.

We also made the change in the table 2 title: “Descriptive characteristics among the different types of CYBA rs4673 and GPx-1 rs1050450 SNPs.”

In line 215 is table 1– it should be table 2.

Response:

Thank you for pointing out this typo, we have corrected.

Moreover, I suggest unifying the units. In tables 1 and 2, the Authors used: Blood Pb - (μg/dL), but in the text, they wrote mcg/dL. The same with P-value (there is P, p, or P value, for example, in and under tables 1 and 2). The writing style must be the same throughout the paper.

Response:

Thank you for pointing out this typo, we have corrected.

In table 3, there are: β and (SE) in the caption – these symbols/shortcuts are not explained.

Response:

Thank you for the comment. β (SE) meant regression coefficient and standard error. We have put it in the table footnote.

In the Methods section in lines 122-123, there is:

"The amplified products were digested at 37°C or 60°C for 1 hour 122 each using 25-μL PCR-amplified product with 10 units of the restriction enzyme…" -

Please specify what enzyme did you use and what for?

Response:

We are sorry that revised fragment of draft editing error, and thank you for this valuable comment. The paragraph was re-written as “Each real-time PCR was performed with a 10 uL reaction volume mix fluid, containing 5 uL Genotyping Master Mix, 3.87 uL distilled water, 1 uL DNA fluid (10 ng/ uL) and 0.25 uL primer-probe. Amplification reactions were performed using the follow-ing program for total 45 cycles: 50°C for 2 minutes; 92°C for 10 minutes; 95°C for 15 seconds; 60°C for 1 minute.” --- lines 128-134

In conclusion, I consider the research presented to be valuable, but in this form, it cannot be published in the International Journal of Environmental Research and Public Health. I recommend revisions and resubmission of a revised article to the editor.

Response:

We appreciate your constructive comments, which we believe have substantially strengthened our manuscript. We have revised the manuscript according to your suggestions, including an overall revision for improved readability and clarity, and hope that our revisions will render the manuscript suitable for publication in International Journal of Environmental Research and Public Health.

Reviewer 3 Report

Article Review: International Journal of Environmental Research and Public Health.

Title: Interaction of smoking and lead exposure increasing oxidative stress indicators among related genetic polymorphisms.

Summary: The goal of this study was to evaluate susceptibility to oxidative stress in occupational workers exposed to lead who have two genetic polymorphisms in antioxidant genes CYBA and GPX1. In addition, they are also investigating interactions between smoking and lead exposure in oxidative stress responses.

Major Revisions

  1. How do the Pb exposure levels in this population compare to other occupational exposures to Pb? Are these workers highly exposed to lead? What types of occupations were evaluated? More demographic information on occupations would be helpful if that was part of the data that was collected.
  2. Discussion paragraph 2, line 253-259. I do not get the connection between this paragraph and the main conclusions of your manuscript. I would move this down lower in the discussion or removed entirely.
  3. In the Discussion paragraphs 2 and 3, the authors discuss differences in disease outcomes (diabetes, CHD and CVD) associated with oxidative stress. However, I do not see these data presented in the results section. I would create a table with all disease state outcomes stratified based on Pd exposure and smoking status and would add all disease state data compared to oxidative stress biomarkers into your multiple linear regression model.
  4. Did you have an internal control for smoking such as Cotinine? This would add so much to your study to not just have self-reported smoking status, but instead have a continuous variable.
  5. Based on your multiple linear regression analysis you observed no effect on polymorphisms and oxidative stress endpoints. This should be discussed in the discussion. Why do you think you did not observe any effects? Limitations in study? How did you choose these two genes to focus on?
  6. Other major limitations to your study was the fact that the majority of your participants were male which made it difficult to look at sex effects and also that lead levels varied between smoking and nonsmoking groups. More explanation of how these factors would adjusted in your multiple linear regression analysis would be helpful.
  7. Discussion, what is the significance of these findings? It would be good to provide a paragraph in the discussion talking about the significance of this study and how more oxidative stress production with combined lead and smoking exposure could lead to increased disease risk.

Minor Revisions

  1. Abbreviations CHD and CVD not defined in text.
  2. Check that all abbreviation is defined at each use.
  3. Multiple grammatical errors, this manuscript should be edited for English.

Author Response

Article Review: International Journal of Environmental Research and Public Health.

Title: Interaction of smoking and lead exposure increasing oxidative stress indicators among related genetic polymorphisms.

Summary: The goal of this study was to evaluate susceptibility to oxidative stress in occupational workers exposed to lead who have two genetic polymorphisms in antioxidant genes CYBA and GPX1. In addition, they are also investigating interactions between smoking and lead exposure in oxidative stress responses.

Major Revisions

  1. How do the Pb exposure levels in this population compare to other occupational exposures to Pb? Are these workers highly exposed to lead? What types of occupations were evaluated? More demographic information on occupations would be helpful if that was part of the data that was collected.

Response:

Thank you for your insightful opinion. We have gone through all records of the occupational exposures and added demographic information. Due to law of information confidential, we cannot release more about kinds of occupations. However, the research revealed the blood Pb concentrations. It is helpful to differentiate the Pb workers and non-Pb workers, which the blood Pb concentrations almost less than 5 ug/dL for non-Pb workers in Taiwan.

  1. Discussion paragraph 2, line 253-259. I do not get the connection between this paragraph and the main conclusions of your manuscript. I would move this down lower in the discussion or removed entirely.

Response:

We would like to thank for the comment. The paragraphs 2 and 3 were re-written as a new paragraph 2 to discuss the indicator oxLDL/LDL. Please refer to the revised manuscript.

  1. In the Discussion paragraphs 2 and 3, the authors discuss differences in disease outcomes (diabetes, CHD and CVD) associated with oxidative stress. However, I do not see these data presented in the results section. I would create a table with all disease state outcomes stratified based on Pd exposure and smoking status and would add all disease state data compared to oxidative stress biomarkers into your multiple linear regression model.

Response:

Thank you for your pertinent comments that have helped us to improve the quality and readability of our manuscript. The paragraphs 2 and 3 were re-written as a new paragraph 2 to discuss the indicator oxLDL/LDL. Please refer to the revised manuscript.

  1. Did you have an internal control for smoking such as Cotinine? This would add so much to your study to not just have self-reported smoking status, but instead have a continuous variable.

Response:

Thank you for your insightful opinion. But we did not have the internal control data of smoking, such as cotinine regretfully. However, we have put this point in our limitation of this manuscript, and wish could initiate this measurement in the next study. (lines 337-340)

  1. Based on your multiple linear regression analysis you observed no effect on polymorphisms and oxidative stress endpoints. This should be discussed in the discussion. Why do you think you did not observe any effects? Limitations in study? How did you choose these two genes to focus on?

Response:

Thanks for the comments. In our regression models, we did not find the interaction effect of polymorphisms and blood Pb or polymorphisms and smoking. We have discussed in the revised manuscript, discussion paragraphs 4 and 5, lines 295-303, and lines 318-324.

  1. Other major limitations to your study was the fact that the majority of your participants were male which made it difficult to look at sex effects and also that lead levels varied between smoking and nonsmoking groups. More explanation of how these factors would adjusted in your multiple linear regression analysis would be helpful.

Response:

Thank you for this valuable comment. This is indeed a limitation of the study, and we stated this aspect the limitation part of Discussion. (Lines 341-345)

  1. Discussion, what is the significance of these findings? It would be good to provide a paragraph in the discussion talking about the significance of this study and how more oxidative stress production with combined lead and smoking exposure could lead to increased disease risk.

Response:

Thanks for the comment. The paragraph 1 of the discussion and conclusion of the revised manuscript are the significance of these findings. We think smoking cessation and reducing Pb exposure are the main issues in public health, other than susceptible genotypes.

Minor Revisions

  1. Abbreviations CHD and CVD not defined in text.

Response:

Thanks for the comment. We have corrected in the revised manuscript.

  1. Check that all abbreviation is defined at each use.

Response:

Thanks for the comment. We have corrected in the revised manuscript.

  1. Multiple grammatical errors, this manuscript should be edited for English.

Response:

Thanks for the comment. The manuscript has been edited by a native speaker (from the U.S.A.), our English teacher and editing service in the University. We hope the revised manuscript are more readable, clarity.

Round 2

Reviewer 1 Report

The article is now ready for publication.

Author Response

Comments and Suggestions for Authors: The article is now ready for publication.

Response:

We are appreciated very much!

Reviewer 2 Report

The authors took all comments into account, to my satisfaction. They corrected the article and improved it significantly.

However, I still have a suggestion to correct a method section. It is still confusing.

In the presented version, after corrections in lines 118-124, it is written that DNA amplification was performed. I think this part may not be deleted from the old version.

Or was preamplification made before real-time PCR? (I doubt that).

Please look carefully at this section.

Moreover, I have noticed in the sentence in line 282/283 expression: “rs1050450 gene” – please use “rs1050450 locus” or just „rs1050450”.

After mentioned above corrections, I recommend this article for publication.

Author Response

We are appreciated very much! Please see the attachment, a point to point response.
